# The Effect of Exoskeletal-Assisted Walking on Spinal Cord Injury Bowel Function: Results from a Randomized Trial and Comparison to Other Physical Interventions

**DOI:** 10.3390/jcm10050964

**Published:** 2021-03-02

**Authors:** Peter H. Gorman, Gail F. Forrest, Pierre K. Asselin, William Scott, Stephen Kornfeld, Eunkyoung Hong, Ann M. Spungen

**Affiliations:** 1Department of Neurology, University of Maryland School of Medicine, Baltimore, MD 21201, USA; 2Division of Rehabilitation Medicine, University of Maryland Rehabilitation and Orthopaedic Institute, Baltimore, MD 21207, USA; 3Kessler Foundation, West Orange, NJ 07052, USA; gforrest@kesslerfoundation.org; 4Department of Physical Medicine and Rehabilitation, Rutgers New Jersey Medical School-Rutgers University, Newark, NJ 07103, USA; 5Spinal Cord Damage Research Center, James J. Peters VA Medical Center, Bronx, NY 10468, USA; Pierre.Asselin@va.gov (P.K.A.); Stephen.Kornfeld@va.gov (S.K.); Eunkyoung.Hong@va.gov (E.H.); Ann.Spungen@va.gov (A.M.S.); 6Department of Rehabilitation and Human Performance, Icahn School of Medicine at Mount Sinai, New York, NY 10029, USA; 7VA Maryland Healthcare System, Baltimore, MD 21201, USA; William.scott5@va.gov

**Keywords:** spinal cord injury, bowel function, exoskeletal walking, constipation

## Abstract

Bowel function after spinal cord injury (SCI) is compromised because of a lack of voluntary control and reduction in bowel motility, often leading to incontinence and constipation not easily managed. Physical activity and upright posture may play a role in dealing with these issues. We performed a three-center, randomized, controlled, crossover clinical trial of exoskeletal-assisted walking (EAW) compared to usual activity (UA) in people with chronic SCI. As a secondary outcome measure, the effect of this intervention on bowel function was assessed using a 10-question bowel function survey, the Bristol Stool Form Scale (BSS) and the Spinal Cord Injury Quality of Life (SCI-QOL) Bowel Management Difficulties instrument. Fifty participants completed the study, with bowel data available for 49. The amount of time needed for the bowel program on average was reduced in 24% of the participants after EAW. A trend toward normalization of stool form was noted. There were no significant effects on patient-reported outcomes for bowel function for the SCI-QOL components, although the time since injury may have played a role. Subset analysis suggested that EAW produces a greater positive effect in men than women and may be more effective in motor-complete individuals with respect to stool consistency. EAW, along with other physical interventions previously investigated, may be able to play a previously underappreciated role in assisting with SCI-related bowel dysfunction.

## 1. Introduction

Spinal cord injury (SCI) is well known to adversely affect bowel function [1,2,3]. Constipation related to slowed colonic transit time is a major issue related to positioning in non-ambulatory individuals and has been specifically demonstrated in SCI [4]. Greater than one third of male participants with SCI reported via a survey that bowel and bladder dysfunction had the most significant effect on life after SCI [5]. Standard bowel management approaches include the manipulation of the diet, oral laxatives and stool softeners, rectal suppositories and enemas, digital rectal stimulation, the use of evacuation equipment, and the timing or scheduling of bowel care [6]. There are some suggestions that a frequent upright posture might help with bowel function. In a Canadian survey study of adults with SCI, 30% of the respondents (38 out of 126) indicated that they participated in prolonged standing (40 min per session, 3 to 4 times per week) in order to improve or maintain health. Of those respondents, 20 out of 38 indicated that bowel and bladder function was one of the main perceived benefits, and 17 indicated that “digestion” was improved [7]. An Australian study [8] explored the specific question as to whether a six-week standing protocol in wheelchair-dependent persons with chronic SCI would improve the time to first stool as well as several other secondary outcomes (the time to complete bowel care, neurogenic bowel dysfunction (NBD) score [9], Cleveland Clinic Constipation Score [10], and St Mark’s Incontinence Score [11]). This was a single-blind, randomized, crossover, controlled study with a four-week washout period, and standing was accomplished through the use of a tilt table for 30 min, five times per week for six weeks. The study demonstrated no effect on the time to first stool, nor any treatment effect on any of the other secondary outcome measures. There was, however, a perception on the part of eight out of the 20 participants that standing “improved” bowel function, although what, exactly, this meant was not reported [8].

In the able-bodied population, it is well established that walking as a form of exercise can enhance bowel motility [12,13]. In a study of inactive middle-aged patients with chronic idiopathic constipation, a 12-week physical activity program, which included brisk walking, statistically reduced total colonic transit time when compared to sedentary controls [14]. This phenomenon may be more related to activity than upright posture, as a study comparing treadmill running, bicycle ergometry and rest in a chair demonstrated improvements in bowel transit time in the two active arms but not the sedentary one [15]. It is therefore reasonable to consider the effect of locomotor interventions on bowel function in those with SCI. In a prospective observational cohort study of locomotor training as well as overground standing and stepping activities in those with motor-incomplete chronic SCI, significant improvement was documented in the sensation of bowel movement, and although not statistically significant, improvements in stool continence occurred in 7 out of 16 individuals with reduced or absent continence at baseline [16]. In a prospective cohort study of seven chronic participants with a range of SCI from C4-through-T5 including both complete and incomplete injuries, who underwent 80 sessions of locomotor training alone, there was a significant decrease in the time required for defecation and a decrease in fecal incontinence as well [17]. Exoskeletal-assisted walking may be another intervention that could potentially improve bowel function in this population [18]. In this prospective single-group case series, ten persons with motor-complete SCI who completed 25–63 sessions of exoskeletal-assisted training over 12 to 14 weeks were provided bowel function surveys at baseline and post-training. These included the total bowel evacuation time per bowel day, frequency of bowel evacuations per week, and Bristol Stool Form Scale (BSS) [19] stool consistency assessments. More than 50% of the participants reported some aspect of improvement in bowel management and/or bowel function. Four participants went on to participate in an additional two months of exoskeletal-assisted walking training, and post-measurements were performed at one-month post-training. All four of these participants reported a decrease in total bowel evacuation time during exoskeletal training [20]. Three out of four reported normalization of stool consistency on the BSS, and three out of four reported the elimination of the need for bowel medications during training, although they required them prior to and after training. These pilot data were encouraging. Therefore, we proposed studying bowel management and function as secondary outcome variables in more detail in the context of a randomized clinical trial of the effects of 36 sessions of exoskeletal-assisted walking in individuals with chronic non-ambulatory SCI.

## 2. Experimental Section

### 2.1. Recruitment

This study was approved by the Institutional Review Boards (IRBs) of three collaborating clinical sites: (1) The James J. Peters VA Medical Center (JJPVAMC), Bronx, NY; (2) The University of Maryland, Baltimore, IRB for the University of Maryland Rehabilitation and Orthopaedic Institute (UM Rehab and Ortho), Baltimore, MD; and (3) The Kessler Foundation (KF), West Orange, NJ. In addition, the Department of Defense Congressionally Directed Medical Research Program, Spinal Cord Injury Research Program (SCIRP) Human Research Protection Office (HRPO) approved the overall study. Several recruitment strategies were employed. Study physicians at each site were the primary source for identifying potential participants. Additionally, IRB-approved flyers and brochures were distributed at each site. Third, some participants self-identified through the clinicaltrials.gov website listing (NCT02314221). All the potential participants were informed about the details and eligibility for the study. The targeted study population was those with chronic SCI (≥6 months) who were non-ambulatory and therefore used wheelchairs for mobility.

### 2.2. Protocol

A three-center, randomized, crossover, controlled clinical trial of exoskeletal-assisted walking (EAW) was designed and implemented in non-ambulatory individuals with chronic SCI (>6 months post-injury). The primary aim of the study was to determine the number of sessions necessary to achieve adequate EAW skills and hypothesized velocity milestones. The mobility component of the study, as well as the detailed eligibility criteria, has been published elsewhere [21]. Briefly, individuals with paraplegia or tetraplegia greater than six months in duration, between 18 and 65 years old, unable to ambulate faster than 0.17 m/s on level ground, wheelchair dependent for mobility, and without any history of concurrent medical or neurologic disease or history of lower extremity fracture within the past two years were eligible for the study. There were no specific inclusion or exclusion criteria that were based specifically on bowel function. As a secondary outcome measure, we investigated whether an EAW intervention would improve bowel function as compared to usual activity (UA).

Individuals were screened using a complete history and physical examination incorporating the following: the International Standards for Neurological Classification of SCI (ISNCSCI) examination to determine the level and completeness of injury (the American Spinal Injury Association Impairment Scale (AIS A to D)) and the ranges of motion at the hips, knees and ankles bilaterally; an Ashworth spasticity examination in the lower extremities; a standing orthostatic tolerance test; and the bone mineral density (BMD) scanning of bilateral knees (proximal tibia and distal femur) and hips (femoral neck and total hip) by dual energy X-ray absorptiometry (DXA).

The eligible participants were randomized within the site to one of two groups for 12 weeks (three months): Group 1 received EAW first, three times per week for 12 weeks, and then crossed over to UA for a second 12 weeks; Group 2 received UA first for 12 weeks and then crossed over to EAW for 12 weeks of training.

Two powered exoskeleton devices were used in this study, namely, the ReWalk™ (ReWalk Robotics, Marlborough, MA) [22,23] and the Ekso GT™ (Ekso Bionics, Richmond, CA) [24]. These powered exoskeletons were chosen because they were the only devices commercially available and Food and Drug Administration (FDA)-approved for use within rehabilitation centers at the time of study development. The ReWalk^TM^ was approved for FDA market clearance in 2014, and the Ekso^TM^, in 2016. The two exoskeletal systems are similar in that the external frame supports the user at the feet, ankles, legs, pelvis and lower trunk. Lofstrand crutches or a walker are required for balance and stability during standing, stepping and walking. Additional information about the exoskeletal training and decision tree for the devices has been published elsewhere [21].

Within the first two sessions, standing balance skills were practiced and achieved prior to progression to walking skills. Walking skills were then initiated utilizing a weight-shifting pattern. Continuous walking resulted from a serial performance of weight shifting. Participants were advanced in their degrees of activity and numbers of steps based on individual progress as determined by the instructing trainer. Missed sessions were added to the end of the 12 weeks to achieve a 36-session total intervention.

The effect of exoskeletal-assisted walking on bowel function was assessed using three instruments: a modified bowel survey modelled after Lynch et al. [25] that we called the 10 Question Bowel Function Survey (10Q), the BSS [19,26], and the short-form item bank for Bowel Management Difficulties from the SCI-Quality of Life instrument (SCI-QOL) [27,28]. These instruments were used three times: at baseline, at crossover, and after the second arm for both the UA and the EAW group. The 10Q consisted of questions felt to be important to the principal investigators in assessing patient-reported bowl management issues, specifically with regard to bowel program satisfaction, the time it took to perform a bowel program, the amount of enema assistance needed, the amounts of oral laxatives and stool softeners used, the frequency of digital stimulation needed, and the frequency of unwanted bowel evacuation episodes. The 10Q has not previously been validated. The BSS provided information about stool consistency. The BSS rates stool consistency from 1 (hard to pass) to 7 (watery liquid), where 4 is the desired medium consistency in persons with upper-motor-neuron bowel dysfunction. It has been validated in the context of other disease entities [26]. The Bowel Management Difficulties item bank from the SCI-QOL instrument consisted of 26 items scored on a five-point Likert scale (possible score range, 26–130). The SCI-QOL scores were standardized on a T-metric, according to a previously published T-score conversion table for SCI-QOL Bowel Management Difficulties [29]. Lower scores indicate greater satisfaction with management. The items included such statements as the following: bowel accidents limited my independence, I worried about performing my bowel program, and I was frustrated by repeated bowel accidents. Validation of the SCI-QOL has been performed [28,29]. The 10Q survey and the BSS scales are available in the Appendix A as is information on how to access the SCI-QOL instrument.

## 3. Results

A total of 50 participants completed the exoskeletal-assisted walking protocol including crossover. Of these, 49 individuals completed the bowel surveys and the BSS at all the time points. Males represented 76% of the participants (*n* = 38). The percent of paraplegic individuals was 72%. The duration of SCI was greater than two years for 52% of the participants, while 48% of the participants were six months to two years out from their injury. Participants with motor-complete injury (American Spinal Injury Impairment Scale (AIS) A and B) represented 62% of the participants, and those with motor-incomplete injury (AIS C and D) made up the remaining 38%. More complete demographic data have been previously presented [21].

The 10Q Bowel Function survey results specifically related to external assistance needed and bowel evacuation times are presented in Table 1. Results from five out of the 10 questions are presented in this table, with the responses from three similar questions detailing the extent of external help needed combined for the purposes of analysis. The five-point scales used in the 10Q survey (see the Appendix A for specifics) were then coded into a binary categorization to allow for qualitative comparisons pre- and post-EAW. In looking at the whole group (regardless of the randomization order), 12% of the participants reported a reduced need for external help, and 24% of the participants reported a reduced evacuation time during each session and across a full week after the 36 sessions of EAW. Analysis of the other five questions did not demonstrate any effect from the EAW or the UA intervention.

The BSS data suggested a qualitative improvement, as the participants reported an improvement (i.e., trend towards a medium stool consistency of 4) after EAW not seen in the UA group. By chance, the UA group at baseline tended to do better than the baseline EAW group, and further improvement in the UA group was not seen (Figure 1). When the BSS grades 5 and 6, representing loose stools, were grouped together, the percentage of loose stools changed from 19.1% pre-EAW to 9.3% post-EAW, whereas there was less of a change with usual activity (19.0% pre- to 15.2% post-UA). In an analysis based on gender as an independent variable, the percentage of men with loose stools decreased with EAW (22.2% pre- to 9.1% post-EAW), but the percentage did not change in women (9.1% pre- to 10.0% post-EAW), although the baseline percentages were different in men versus women. According to an analysis comparing motor-complete versus motor-incomplete cohorts, the EAW intervention reduced the percentage of loose stools from 23.3% to 6.9% (*n* = 31) in the motor-complete participants, whereas there was a slight worsening in the motor-incomplete subgroup (11.8% pre- to 14.3% post-EAW, *n* = 17).

The results from the Bowel Management Item Bank components of the SCI-QOL are presented in Table 2. Overall, for the whole group, there were no significant effects found for changes in patient-reported outcomes for the Bowel Management Difficulties SCI-QOL survey after EAW in comparison to UA. The only statistically significant beneficial preintervention–postintervention change was seen during the EAW phase for the participants who started in the UA-first group. An improvement from 49.5 ± 9.2 to 46.5 ± 9.8 (*p* = 0.028) was noted (a lower score indicating better satisfaction).

We performed several *post hoc* subgroup analyses based on (1) the time since injury, (2) gender and (3) motor-complete (AIS A/B) versus motor-incomplete (AIS C/D) injury. The time since injury analysis was considered important since it was noticed that the more newly injured participants (less than two years since injury) were often still learning to maximize their bowel management. Stratification by the duration of injury (DOI) subcategories for the outcomes of bowel function showed that those persons injured for more than two years demonstrated an improvement trend in the Bowel Management SCI-QOL survey after EAW. By contrast, the more newly injured cohort (DOI < 2 y) did not show improvement. A comparison of the effects of EAW on the SCI-QOL bowel management item bank in men vs. women was performed. For men (*n* = 38), the average score decreased from 50.1 to 47.6 with EAW (*p* = 0.041), but in women, there was an average score increase from 48.5 to 51.2, albeit, non-significant (ns). When those with motor-complete injury were compared with those with motor-incomplete injury, the EAW intervention did not produce any beneficial effect in either group. Surprisingly, however, the usual activity intervention produced a statistically significant preintervention–postintervention change in the motor-incomplete cohort. Comparisons between the changes observed with the EAW and the UA interventions, when examined in total and in all the subgroup analyses, demonstrated no statistically significant differences.

## 4. Discussion

EAW training had a positive effect on about one quarter of the participants for the patient-reported outcomes for bowel function and management. There were also trends towards normalization of stool consistency in the EAW group not seen in the UA group. Men responded better than women to EAW in terms of reductions in loose stools. Those with motor-complete injury responded to EAW in terms of reductions in loose stools, whereas the motor-incomplete group did not. The overall results from the Bowel SCI-QOL batteries did not show a significant improvement in patients’ perceptions of their evacuation management with the EAW intervention. The time-since-injury sub-analysis suggested that those with newer SCI may still be adjusting and becoming competent with their bowel program, thus negating any potential positive effect from the EAW intervention. When the bowel SCI-QOL results were examined by gender, it was noted that men responded to EAW significantly, whereas women did not, although the enrolled women started off scoring slightly better (lower) on this scale (48.5 vs. 50.1). When the results were examined by the presence or absence of motor completeness, the usual activity intervention was associated with a significant improvement, whereas the EAW intervention was not. It is to be noted, however, that in both the EAW and UA groups, the baseline bowel SCI-QOL scores were better in the participants with motor-incomplete (AIS C/D) SCI. This makes intuitive sense, as these individuals at baseline had more intact corticospinal tracts in their spinal cords (as demonstrated by preserved motor control) and, therefore, would be expected to have better bowel control. It is difficult to draw direct conclusions as to the clinical significance of the observed effect of usual activity in the participants with motor-incomplete injury.

The effect of exoskeletal-assisted walking on bowel function in spinal cord injury was not as dramatic as was hypothesized. Our 25-to-63-session EAW pilot investigation had suggested a more robust finding. Indeed, in 10 participants, there were a reported reduction in the time spent having a bowel movement (5/10), fewer bowel accidents (6/10), a decreased frequency of laxative and/or stool softener use over the prior week (7/10), and a reported improvement in overall satisfaction with the bowel program (6/10) [18]. This much larger randomized study did not confirm these findings. Rather, more subtle improvements were noted, and only one out of the two preintervention–postintervention comparisons demonstrated a significant improvement with the EAW intervention. When all of the preintervention–postintervention EAW data were evaluated together, irrespective of the order of the EAW intervention, the statistical significance of the effect was lost.

With regard to the self-reported outcomes, the improvement in approximately one quarter of the participants in bowel evacuation time did confirm prior pilot data. This contrasts with the lack of improvement in bowel evacuation time found by Kwok et al. in a study of standing alone discussed previously [8]. One might therefore postulate that the actual activity associated with exoskeletal walking rather than just the upright posture alone was the causative agent leading to improved transit times. This might be related to the trunk exertion needed to shift weight during stepping, although further work would be needed to confirm this.

The Bristol stool form scale perhaps provided the most intriguing results. Normalization of the stool consistency (i.e., towards the middle category 4 on the 1–7 scale) after EAW walking was noted and appeared not to have occurred after UA. The baseline values prior to UA included a number of category 4 reports, which likely made it more difficult to make a comparison statistically significant. Nonetheless, this effect is a relevant one with regard to the risk of incontinence in those with spinal cord injury. The fact that the motor-complete subgroup responded to EAW in terms of a reduction in loose stools is likely due to the fact that they were worse off at baseline, and achieving overground walking through the robotic intervention likely represented a more dramatic change in physiology than in the motor-incomplete group.

The results provided here lend support to the idea that upright overground activity (i.e., walking) and not just standing may have a beneficial effect on bowel function in those with chronic spinal cord injury, but the size of the effect was not as profound as hypothesized. Nonetheless, the collection of these secondary outcome data was not excessively burdensome and likely should be included in the future study of any type of mobility intervention in people with spinal cord injury.

## 5. Conclusions

A 36-session exoskeletal-assisted walking program implemented over three months in non-ambulatory persons with spinal cord injury provided some, albeit limited, improvement in several measures of bowel function when compared to a usual activity control group. The most notable improvement (i.e., reduction) was seen in average bowel evacuation time and in a trend to the normalization of bowel form consistency. The degree of subjective improvement as determined by quality-of-life bowel survey instruments may be, in part, related to the time since injury, and exoskeletal-assisted walking may have a predilection to benefit men more than women. The bowel survey quality-of-life results at baseline were confirmed to be better in motor-incomplete persons than those with motor-complete injuries, but individuals with motor-complete injury may improve more readily with an EAW intervention in terms of stool consistency.

## Figures and Tables

**Figure 1 jcm-10-00964-f001:**
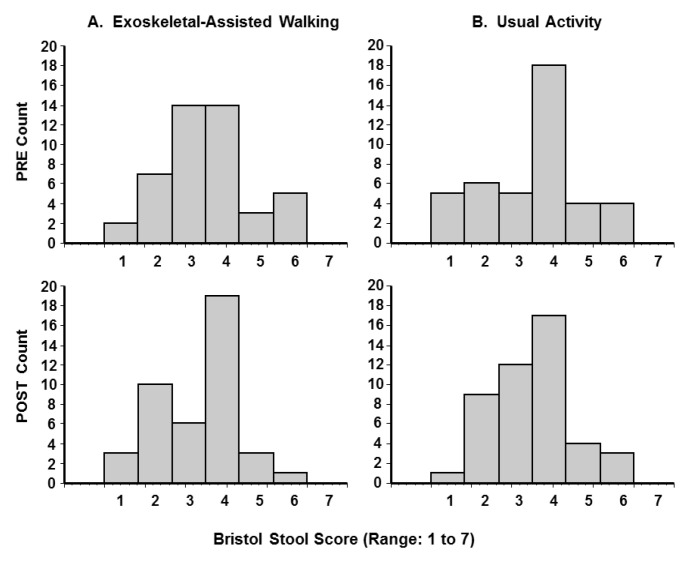
Bristol Stool Form Scale (BSS) results. Frequency distribution of pre- and post-exoskeletal-assisted walking and pre- and post-usual activity. The top row represents preintervention, and the bottom row, postintervention data. Larger values on the 1 to 7 BSS represent looser stools. Details of the BSS can be found in the Appendix A.

**Table 1 jcm-10-00964-t001:** Selected patient-reported outcomes from the 10Q Function Survey. Results from three of the 10Q questions were combined in the first category in the table in order to represent the degree of external assistance needed in order to accomplish a bowel evacuation. The other two categories represent results from one individual question from the 10Q. For each category, a new binary response was created by lumping responses in order to qualitatively compare pre -and post-EAW results. For the purpose of the results in the last column, improvement was defined as either reduction in external assistance or reduction in bowel evacuation time.

Category	Frequency Reported	Pre-EAW	Post-EAW	Percent of Participants Improved
Enema, oral medication and/or manual digital stimulation needed for each bowel evacuation in the past week	Never to a few times	57%	63%	12%
	Most to every time	41%	35%	
Average bowel evacuation time needed per bowel day during the past week	5 to 60 min	80%	92%	24%
	>1 to 3 h	18%	6%	
Average bowel evacuation time needed in the past week	1 to 6 h	80%	92%	24%
	>6 to 8 h	16%	6%	

**Table 2 jcm-10-00964-t002:** Bowel Management Item Bank from the Spinal Cord Injury Quality of Life (SCI-QOL).

Category	Exoskeletal-Assisted Walking (EAW) Phase (*n* = 50)				Usual Activity (UA) Phase (*n* = 49)				EAW vs. UA Diff (*p* Value)
	Pre	Post	Paired T-Test (*p*)	Pre–Post Diff	Pre	Post	Paired T-Test (*p*)	Pre–Post Diff	
**All (*n* = 49)** **(± SD)**	49.7 (8.7)	48.4 (9.2)	0.207	−1.3 (7.1)	50.8 (8.6)	49.3 (9.2)	0.071	−1.5 (5.8)	0.88
**(range)**	36.1–79.4	36.1–74.4			36.1–75.0	36.1–79.4			
**EAW first** **(*n* = 24)**	49.9 (8.4)	50.4 (8.3)	0.292	0.5 (7.4)	50.4 (8.3)	49.1 (9.5)	0.292	−1.3 (5.8)	0.46
**UA first** **(*n* = 25)**	**49.5 (9.0)**	**46.5 (9.8)**	**0.028**	**−3.0 (6.5)**	51.2 (8.9)	49.5 (9.1)	0.145	−1.8 (5.8)	0.54
**DOI > 2.0 y** **(*n* = 26)**	51.0 (9.5)	48.8 (10.2)	0.144	−2.2 (7.3)	52.0 (9.4)	50.9 (9.6)	0.299	−1.1 (5.4)	0.60
**DOI ≤ 2.0 y** **(*n* = 23)**	47.9. (7.6)	47.5 (8.3)	0.823	−0.4 (6.9)	49.1 (7.5)	47.0 (8.5)	0.143	−2.0 (6.2)	0.51
**Male** **(*n* = 38)**	**50.1 (9.1)**	**47.6(9.8)**	**0.041**	**−2.5 (7.1)**	50.8 (9.0)	49.5 (9.3)	0.109	−1.3 (5.0)	0.49
**Female** **(*n* = 11)**	48.5 (6.5)	51.2 (5.3)	0.108	2.8 (4.9)	50.8 (6.3)	48.6 (8.3)	0.398	−2.1 (7.7)	0.23
**Complete** **(*n* = 31)**	51.6 (9.3)	49.9 (9.6)	0.243	−1.7 (7.8)	52.1 (9.1)	51.4 (9.7)	0.489	−0.7 (5.7)	0.65
**Incomplete** **(*n* = 18)**	46.5 (6.6)	45.9 (8.1)	0.642	−0.6 (5.8)	**48.6 (7.4)**	**45.7 (7.0)**	**0.047**	−2.9 (5.8)	0.32

Note that a lower score represents a better outcome. *n* = 49 for the usual activity (UA) group, as one participant in UA-first group was lost to follow-up after the UA arm for this outcome; values in parentheses are ± standard deviation; EAW = exoskeletal-assisted-walking-first group; UA = usual-activity-first group; DOI = duration of injury; y = years; Diff = the difference from pre- to postintervention; shaded area indicates statistically significant value from pre- to postintervention. Bold type font represents statistically significant changes. Thicker lines are placed between sets of rows for easier comparisons.

## Data Availability

The data presented in this study are available on request from the corresponding author. The data are not publicly available due to restrictions based on Department of Veterans Affairs regulations.

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
