# Peer review of "The Effect of Exoskeletal-Assisted Walking on Spinal Cord Injury Bowel Function: Results from a Randomized Trial and Comparison to Other Physical Interventions"

_jcm, 2021, doi:10.3390/jcm10050964_

Round 1

Reviewer 1 Report

Ms. No.: JCM-1077496

Title: Exoskeletal-Assisted Walking Effect on Spinal Cord 2 Injury Bowel Function: Results from a Randomized 3 Trial and Comparison to other Physical Interventions

The aim of this study was to determine the reduction in neurogenic bowel symptoms following a 36-week protocol of exoskeleton assisted walking or usual activity. Bowel function was assessed using 3 patient compiled questionnaires (SCI-QOL, Bristol Stool scale and a proprietary 10 question function survey). The study was performed as a randomized multi-center crossover-controlled clinical trial.

The main results reported some measure of improved patient satisfaction following training. The use of stimulants decreased following EAW as did total time to perform bowel program. However, these effects were below expectations of the investigators and may be considered rather modest in relation to the cost and availability of exoskeletal-assisted walking.

Nonetheless, this report offers an insight to understand the benefits, and limitations, of upright posturing and exoskeleton-assisted ambulation on neurogenic bowel; a profound comorbidity of SCI. The anatomical orientation of the human lower bowel is adapted to the challenges of bipedal locomotion. Returning the SCI individual to this orientation may non-invasively promote eliminative function. The results will be of value to physiatrists and rehabilitation teams, but additional data could have (presumably) been made available to improve understanding of these benefits.

Introduction:

Minor - The introduction is appropriately detailed with regard to the rationale for pursing the intervention study. There may be a typographical error on line 77 (reduced vs reduces).

Patient recruitment and participation:

Moderate – The inclusion/exclusion criterea appear to be lacking in section 2.1

Results:

Moderate – The gender distribution of participants is satisfactory, however, data was presented without regard to gender. Gastrointestinal complaints are more frequently reported in able bodied females and it would be of interest if there were any differences between SCI males and females.

Moderate – For the participants reporting improvements. Were these improvements experienced rapidly by the EAW or was data collected only for the terminal (12-week) week of the intervention?

Moderate – What portion of those that benefitted from the EAW were in the motor incomplete category?

Discussion:

No concerns

Author Response

The authors appreciate the comments presented by the reviewer.  Individual comments and suggestions are addressed below:

1) The typographical error on line 77 has been corrected

2) An overview of the inclusion/exclusion criteria has been added to section 2.2. The complete set of inclusion and exclusion criteria are provided in reference 21.

3) We appreciate the reviewer’s suggestion to further investigate the effect of gender on the results.  Indeed, by doing so we uncovered a difference in improvement in the firming up of stool consistency in men when compared to women. We also found an unexpected difference between men and women in terms of the effect of EAW on bowel quality of life measures. This information is added to table 2 and is discussed in the results and discussion. I was going to bring in your comment re able-bodied women have more GI complaints than men, but I could not find a reference for that for the lower GI tract, only for the upper GI tract (Haag S. Aliment Pharmacol Ther 2011; 33: 722–729).

4) Because of gaps in the data collected during the EAW and UA interventions, and not just at the end of the 12-week intervention, there is not sufficient information available to assess the speed of improvement, and therefore we have chosen not to present this incomplete information.

5) We also analyzed the differences in changes in stool consistency in the motor complete and motor incomplete subjects as well as in SCI-QOL data. This information is presented in the Results and Discussion sections and the SCI-QOL data is also added to figure 2. 

It should be noted that during the process of reevaluation of the BSS and SCI-QOL data with respect to gender and motor completeness, minor errors in determination of averages and standard deviations were discovered, and therefore other changes were made in the manuscript, specifically in table 2.   The overall manuscript was proofread again, and a few additional corrections and one author’s email address change was made.

Reviewer 2 Report

The research on this topic is very interesting. I liked the study very much. I just have one general question.

The management of bowel function after spinal cord injury is very critical for paraplegics with complete spinal cord injury. For those paraplegics with incomplete spinal cord injury, the management of such function can be very easy as the able-bodied individuals. Considering such variation in the management of the bowel function, I was wondering if the level of difficulty of the management was uniform amongst the participants. Did you recruit paraplegics with moderate to high difficulty with bowel management?

Author Response

The authors appreciate the positive feedback from the reviewer. Indeed, bowel dysfunction and the burden that it places on persons with spinal cord injury can vary considerably.  There is not uniformity to this problem, as is demonstrated to the wide range of values noted on the bowel management items bank from the SCI-QOL (found near the top of Table 2).  We did not specifically recruit individuals with a high degree of bowel management difficulty. In order to be clear about this, a statement to this effect is added to section 2.2 in the expanded inclusion/exclusion criteria discussion.